

# Metamorphosis of the invasive ascidian *Ciona savignyi*: environmental variables and chemical exposure

Patrick L. Cahill, Javier Atalah, Andrew I. Selwood and Jeanne M. Kuhajek

Cawthron Institute, Nelson, New Zealand

## ABSTRACT

In this study, the effects of environmental variables on larval metamorphosis of the solitary ascidian *Ciona savignyi* were investigated in a laboratory setting. The progression of metamorphic changes were tracked under various temperature, photoperiod, substrate, larval density, and vessel size regimes. Metamorphosis was maximised at 18 °C, 12:12 h subdued light:dark, smooth polystyrene substrate, and 10 larvae mL$^{-1}$ in a twelve-well tissue culture plate. Eliminating the air-water interface by filling culture vessels to capacity further increased the proportion of metamorphosed larvae; 87 ± 5% of larvae completed metamorphosis within 5 days compared to 45 ± 5% in control wells. The effects of the reference antifouling compounds polygodial, portimine, oroidin, chlorothalonil, and tolylfluanid on *C. savignyi* were subsequently determined, highlighting (1) the sensitivity of *C. savignyi* metamorphosis to chemical exposure and (2) the potential to use *C. savignyi* larvae to screen for bioactivity in an optimised laboratory setting. The compounds were bioactive in the low ng mL$^{-1}$ to high µg mL$^{-1}$ range. Polygodial was chosen for additional investigations, where it was shown that mean reductions in the proportions of larvae reaching stage E were highly repeatable both within (repeatability = 14 ± 9%) and between (intermediate precision = 17 ± 3%) independent experiments. An environmental extract had no effect on the larvae but exposing larvae to both the extract and polygodial reduced potency relative to polygodial alone. This change in potency stresses the need for caution when working with complex samples, as is routinely implemented when isolating natural compounds from their biological source. Overall, the outcomes of this study highlight the sensitivity of *C. savignyi* metamorphosis to environmental variations and chemical exposure.

## INTRODUCTION

The Pacific transparent ascidian *Ciona savignyi* Herdman (Cionidae) is a Japanese native with an expanding invasive range that currently includes Argentina, British Columbia, California, New Zealand, Puget Sound, and Spain (*Fofonoff et al., 2003*; *Lambert & Lambert, 1998*; *Smith, Cahill & Fidler, 2010*). *Ciona savignyi* is recognised as a problematic biofouling organism; it reproduces rapidly in invaded environments and can dominate man-made and natural substrates (e.g., *Cohen et al., 1998*; *Zvyagintsev, Sanamyan & Kashenko, 2007*). This hermaphroditic species can spawn year-round in temperate regions (*Nomaguchi et al., 1997*; P Cahill, pers. obs.), with each individual releasing hundreds or thousands of eggs to be fertilised in the water column (*Hendrickson et al., 2004*; *Jiang & Smith, 2005*).

Corresponding author
Patrick L. Cahill,
Patrick.Cahill@cawthron.org.nz

Any attempt to counteract the invasive tendencies of *C. savignyi* requires an in-depth understanding of this species life-history characteristics. A key step in the biofouling process is the transition from free-swimming larva to sessile adult (*Pawlik, 1992*). Larvae must contact a suitable surface upon which to settle, choose to attach, and then undergo a complex series of morphological changes to form established juveniles. Marine larvae typically respond to a range of environmental and con-specific cues, with a high degree of variability in responses between species (*Jackson et al., 2002*; *Rodriguez, Ojeda & Inestrosa, 1993*). Little information is available on the extrinsic factors that stimulate *C. savignyi* to metamorphose; identifying these cues will facilitate the development of targeted treatment technologies and mitigation techniques.

In particular, the metamorphic process may be sensitive to exposure to chemical compounds. It has been shown that *C. savignyi* metamorphosis is inhibited by the natural antifouling agent polygodial (*Cahill & Kuhajek, 2014*), with other natural (e.g., oridin or portimine; *Selwood et al., 2013*; *Tsukamoto et al., 1996*) and synthetic (e.g., chlorothalonil and tolyfluanid; *Voulvoulis, Scrimshaw & Lester, 1999*) compounds also likely to be effective against *C. savignyi* metamorphosis. Screening for effects on *C. savignyi* metamorphosis has potential to identify targeted lead compounds, including both known compounds and novel natural compounds. In the case of novel natural compounds, initial screening and isolation typically involves working with complex biological extracts, with potential for interactions between constituents of the extract (*Colegate & Molyneux, 2007*). Understanding how environmental factors influence metamorphosis will improve our ability to reliably quantify antimetamorphic effects.

In addition to the interest in *C. savignyi* as a marine invader, this species has been increasingly studied as a model organism for developmental biology (*Corbo, Di Gregorio & Levine, 2001*; *Sasakura et al., 2012*; *Satoh, 2003*). Ascidians occupy an intriguing evolutionary position as sister clade to the vertebrates (*Lemaire, 2011*; *Satoh & Levine, 2005*; *Schubert et al., 2006*), meaning they a can afford insights into developmental biology in general. Embryos and larvae of *C. savignyi* can be produced in large numbers in the lab, the latter undergoing a defined progression of metamorphic changes to form established juveniles within 7 days (*Cirino et al., 2002*; *Hendrickson et al., 2004*; *Kourakis, Newman-Smith & Smith, 2010*). Many studies have examined the intrinsic determinants driving the progression from egg, to free-swimming larvae, to sessile adult in *C. savignyi* (e.g., *Imai, Satoh & Satou, 2002a*; *Imai, Satou & Satoh, 2002b*; *Kimura, Yoshida & Morisawa, 2003*). Identifying extrinsic conditions that stimulate ascidian larvae to settle and metamorphose will also provide a context for these molecular and biochemical investigations (*Morse, 1990*).

In this study, the effects of extrinsic environmental factors on *C. savignyi* metamorphosis were examined in a laboratory setting. Variables investigated included temperature, photoperiod, substrate, larval density, and vessel size. The most susceptible stages of the larval settlement and metamorphosis processes were identified, providing a relevant experimental end-point for investigating the effects of reference antifouling compounds on the larvae.

## MATERIAL AND METHODS

### Culture and spawning

Adult *C. savignyi*, collected from the underside of pontoons at Nelson Marina, Nelson, New Zealand, were housed in water lily baskets suspended in 10-L glass aquaria for up to three weeks. Aquaria were supplied with 10 L seawater h$^{-1}$ as part of a 1000-L recirculating system held at $18 \pm 1$ °C, $34 \pm 1$ psu, and $300 \pm 50$ mV ORP; constant full-spectrum florescent light prevented premature spawning. Daily, the flow of water to the aquaria was cut off for 3 h while *C. savignyi* were fed 250 mL of an $8-9 \times 10^6$ cells mL$^{-1}$ *Isochrysis galbana* Parke culture. Three gravid individuals with densely packed egg and sperm ducts were spawned according to *Cirino et al. (2002)*. Ventral incisions were made to expose the egg and sperm ducts. The egg duct of each individual was pierced with a Pasteur pipette and the eggs transferred to a glass Petri dish (90 mm dia., 68 mL vol.) filled with 20 mL of 0.3-µm filtered and UV-sterilized seawater (FSW). Sperm were then harvested and transferred to a glass Petri dish (90 mm dia., 68 mL vol.) containing 50 mL FSW. Each dish of eggs received eight drops of sperm suspension from each of the two other individuals. After incubating at 18 °C for 1 h, fertilized eggs were strained through a 20-µm sieve, rinsed three times with 25 mL reconstituted seawater (RSW; $33 \pm 0.5$ psu; Red Sea Salt, Red Sea Aquatics, Cheddar, UK), transferred to a glass Petri dish along with 25 mL RSW, and held at 18 °C for 18 h to hatch. Hatched larvae were transferred to conical flasks and diluted with RSW as necessary to yield desired larval densities.

### Temperature, photoperiod, and substrate

The temperature, photoperiod, and substrate preferences of larvae were assessed in three separate experiments. Firstly, the effects of exposing larvae to 15, 18, 21, and 25 °C were evaluated. For the photoperiod experiment, larvae were exposed to the following lighting regimes: 24:0, 12:12, 0:24 h intense ($100 \pm 10$ µmol m$^{-2}$ s$^{-1}$) or subdued ($10 \pm 5$ µmol m$^{-2}$ s$^{-1}$) light:dark. Substrate type (polystyrene, acrylic, or glass) and texture (smooth or rough) were assessed using unlined wells (polystyrene) or wells lined with acrylic discs (35-mm diameter) or glass cover slips (35-mm diameter, Gerhard Menz GmbH, Saarbrücken, GE). Untreated unlined wells, acrylic discs, and glass cover slips comprised smooth treatments. Sandblasted acrylic and sanded (800-grit sandpaper) glass and polystyrene comprised rough treatments; it should be noted that these surface treatments likely resulted in different feature sizes for each of the three substrate types and that the walls of the wells remained untreated in all cases.

All three experiments (temperature, photoperiod, and substrate) were performed in six-well tissue culture plates (Corning® Costar®, Corning Inc., Corning, NY; 36 mm dia.; 17.5 mL vol.) filled with 10-mL aliquots of 2.5 larvae mL$^{-1}$ RSW. Default parameters were 18 °C, 12:12 h subdued light:dark, and unlined smooth wells; three replicates were performed in all cases ($n = 3$). Culture waters were renewed with RSW every other day and settled larvae were scored according to metamorphic stage (Fig. 1) after 1, 3, 5, and 7 days. Unattached larvae or larvae adhered to the meniscus were not counted. A proxy for daily metamorphic progress was calculated based on the number of larvae counted at each

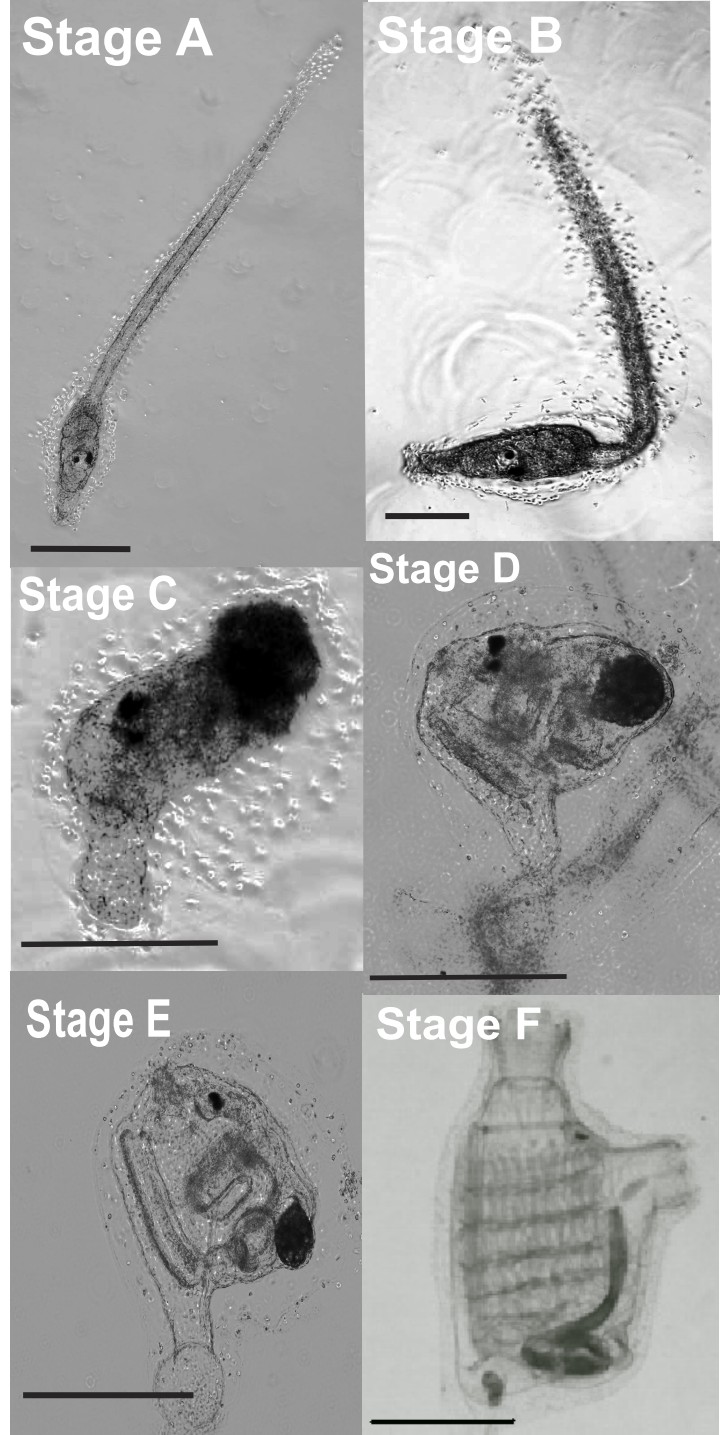

**Figure 1** **Stages of *Ciona savignyi* larval development.** At the moment of settlement, metamorphosis is triggered (Stage A). Tail resorption is completed in a short time (Stage B). The outer tunic layer is cut off (Stages C & D), and then two lateral siphons and a pair of functional stigmata develop (Stage E). The number of stigmata on each side of the branchial basket increases (Stage F), completing metamorphosis. Scale bars = 300 μm.

developmental stage as follows:

$$Metamorphic\ progress = (Stage\ A\ larvae \times 1) + (Stage\ B\ larvae \times 2) + (Stage\ C\ larvae \times 3)$$
$$+ (Stage\ D\ larvae \times 4) + (Stage\ E\ larvae \times 5) + (Stage\ F\ larvae \times 6).$$

## Vessel size, larval density, and fill volume

Vessel size was assessed in conjunction with larval density. Polystyrene Petri dishes (LabServ, Thermo Fisher Scientific, Melbourne, AU; 90 mm dia.; 68 mL vol.), six-well plates, and twelve-well plates (Corning® Costar®; 23 mm dia.; 6.5 mL vol.) were filled with 25-, 10-, or 5-mL aliquots of larval suspension, respectively. Four larval densities (2.5, 5, 10, and 15 larvae mL$^{-1}$) were tested in each of the three vessel sizes, yielding twelve vessel size × larval density treatments in total ($n = 3$). Vessels were held at 18 °C, 12:12h subdued light to dark and culture waters were renewed every other day. After 5 days, the number of larvae that had completed stage E of metamorphosis were counted (Fig. 1).

Fill volume was assessed using twelve-well plates and a larval density of 10 larvae mL$^{-1}$. Treatment wells were filled to capacity with 7.1 mL of larval suspension so that the seawater contacted the underside of the lid, eliminating the air-water interface. Controls wells were filled with 5 mL of larval suspension ($n = 3$). The temperature, photoperiod, and experimental end-point matched those used for the vessel size and larval density experiments but culture waters were not renewed for the duration of this experiment.

## Carrier solvent and reference compounds

The sensitivity of larvae to two common carrier solvents was assessed in twelve-well plates with 7.1 mL aliquots of 10 larvae mL$^{-1}$ RSW; held at 18 °C, 12:12 h subdued light to dark. Concentrations of ethanol and dimethyl sulfoxide (DMSO) evaluated were 0.05, 0.1, 0.25, 0.5, 1, 2.5, 5, 10, 20, and 40 μL mL$^{-1}$; controls contained no solvent ($n = 3$). Larvae that had completed stage E of metamorphosis were counted after 5 days.

Five reference compounds, polygodial (ENZO Life Sciences, Farmingdale, NY, USA), portimine (Cawthron Natural Compounds, Nelson, NZ), oroidin (ENZO Life Sciences), chlorothalonil (Sigma-Aldrich, St Louis, MO, USA), and tolylfluanid (Sigma-Aldrich), were screened using the protocol outlined for carrier solvent. With the exception of polygodial (1, 2.5, 5, 10, 15, 20, 40, 80 ng mL$^{-1}$) and portimine (0.05, 0.1, 0.5, 1, 5, 10, 20 ng mL$^{-1}$), concentrations tested against the larvae were 0.001, 0.005, 0.01, 1, 5, and 10 μg mL$^{-1}$. Stock solutions prepared in 20% (v/v) ethanol (polygodial), RSW (portimine), or 20% (v/v) DMSO (oroidin, chlorothalonil, tolylfluanid) were added to wells to yield the desired test concentrations; control wells contained solvent only ($n = 3$).

Polygodial treatments (5, 10, and 50 ng mL$^{-1}$; $n = 3$) were subsequently included in eight independent experiments performed over a 1-year period, allowing repeatability (i.e., intra-run variability) and intermediate precision (i.e., inter-run variability) to be estimated. In a separate experiment assessing the potential for interactions among complex mixtures of natural compounds, the potency of pure polygodial was compared to that of polygodial enriched with an environmental extract. The extract was produced by eluting 10 L seawater that ~400 g of green-lipped mussels, *Perna canaliculus* Gmelin, had been

cultured in for 1 day through a 20-cm$^3$ column of Diaion HP20 resin (Mitsubishi Chemical, Tokyo, JP). The column was flushed with 50 mL ethanol, and the resulting extract dried and re-suspended in 1 mL 20% (v/v) ethanol. Polygodial stock solutions prepared in 1 mL of 20% (v/v) ethanol or 1 mL of extract were added to wells to yield final polygodial concentrations of 2.5, 5, 10, and 50 ng mL$^{-1}$ ($n = 3$).

## Data analyses

Temperature, photoperiod, and substrate data were analyzed using one-way linear mixed-effects modelling (*Bolker et al., 2009*), with metamorphic progress ('Temperature, photoperiod, and substrate') as the response variable, treatment as a fixed factor, and time as a continuous covariate. Replicate was included as a random effect to account for the repeated-measures experimental design. Reported *p*-values are based on the *t* distribution of the ratios between the estimates and their standard errors (*Pinheiro & Bates, 2006*). Principal response curves (PRC), a redundancy analysis for multivariate responses in repeated-measures design (*Van den Brink & Ter Braak, 1998*; *Van den Brink & Ter Braak, 1999*), were used with 999 permutations to identify the metamorphic stages that were driving treatment effects (>0.5 signifies strong treatment effect). Mean absolute PRC coefficients for the number of larvae in each metamorphic stage (A–F) at each time point were calculated by averaging the values from the temperature, photoperiod, and substrate datasets. The effects of larval density and vessel size were assessed using a two-way factorial ANOVA with the number of Stage E larvae on day 5 as the response variable. The effect of fill-volume on the number of larvae reaching Stage E within 5 days was tested using a Student's *t*-test, and carrier solvent using one-way ANOVA followed by Dunnett's post-hoc test. Dose-response curves were plotted for the reference compounds using four parameter logistic curve fitting, and the corresponding concentrations that reduced the number of larvae reaching Stage E by 50% relative to blank controls ($EC_{50}$) calculated (*Kuo, Mitchell & Tuerke, 1993*). Repeatability (relative bias) and intermediate precision for the replicate polygodial experiments were calculated in accordance with *USP <1033 >(2012)*. Analyses were performed using R 2.13.1 (*R Development Core Team, 2015*) and SigmaPlot 11.0 (*Systat Software Inc, 2015*).

# RESULTS

Overall, larvae reached stages A and B of metamorphosis within 1 day, while stage C took $1.2 \pm 0.06$ days, stage D $3.1 \pm 0.02$ days, stage E $4.3 \pm 0.03$ days, and stage F $6.6 \pm 0.04$ days.

## Temperature, photoperiod, and substrate

Larvae responded to variations in temperature, photoperiod, and substrate. Daily metamorphic progress was comparable for larvae reared at 15, 18, or 21 °C but hindered at 25 °C ($t = -4.6$, $p < 0.001$; Fig. 2A). Metamorphic progress was enhanced relative to the other photoperiod regimes evaluated when exposed to 12:12 h subdued light:dark ($t = 2.1$, $p = 0.04$), and slowed by exposure to constant intense light ($t = 3.0$, $p = 0.004$; Fig. 2B). Metamorphic progress was reduced when the substrate was smooth acrylic

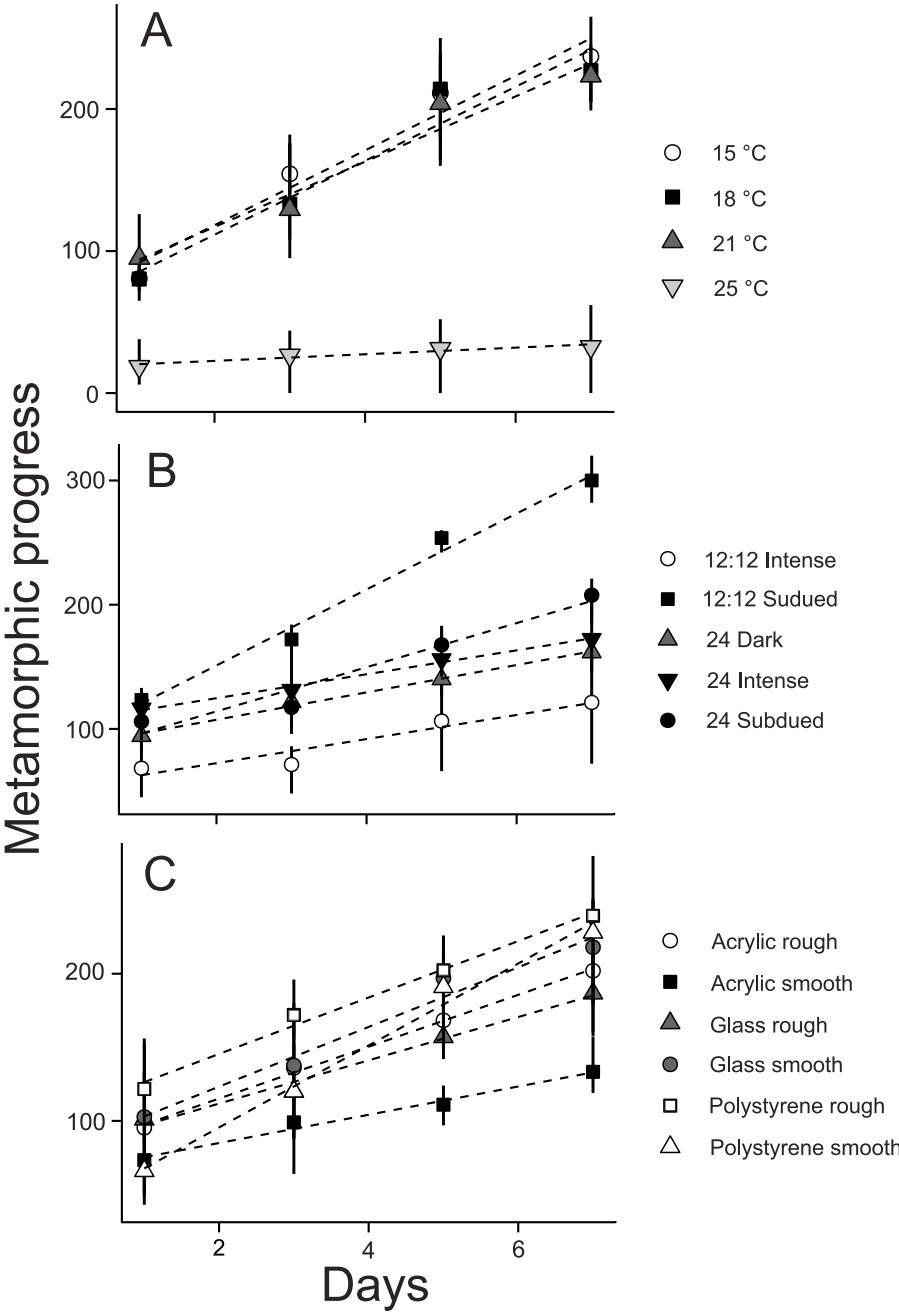

**Figure 2** **Effects of (A) temperature, (B) photoperiod, and (C) substrate on *Ciona savignyi* larvae.** Metamorphic progress was calculated based on the sum of the number of individuals reaching each metamorphic stage converted to numerals (A = 1, B = 2, C = 3, D = 4, E = 5, F = 6). Values are means ($n = 3$) ± 95% confidence interval.

($t = -3.5, p = 0.04$; Fig. 2C) but there were no discernable differences between the other substrate types tested. The statistical differences observed for the temperature (PRC, $F = 10.1, p = 0.005$), photoperiod (PRC, $F = 8.4, p = 0.005$), and substrate (PRC, $F = 1.9, p = 0.005$) datasets were driven by the latter stages of metamorphosis. Mean

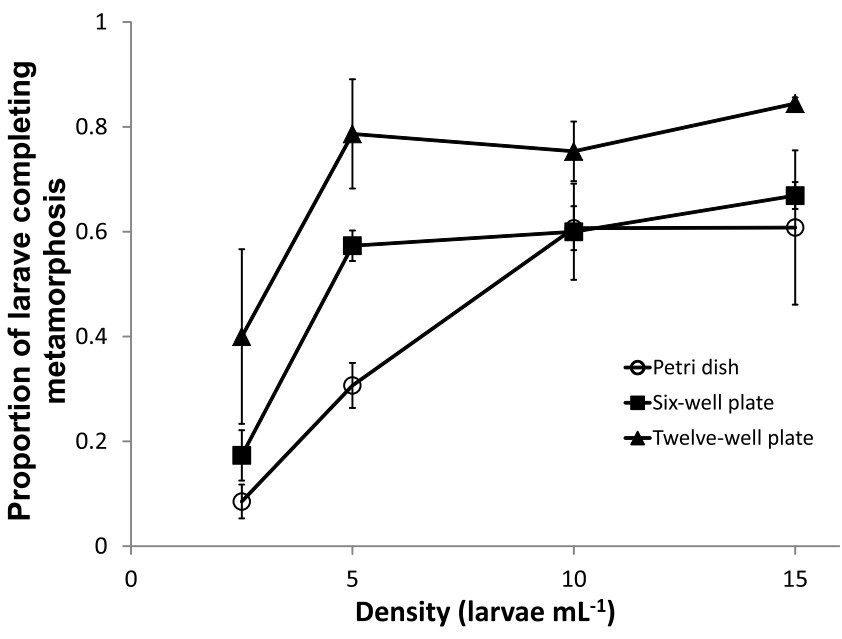

**Figure 3  Proportion of *Ciona savignyi* larvae completing metamorphosis.** Proportion of *Ciona savignyi* larvae completing Stage E of metamorphosis within 5 days for various combinations of larval density and vessel size. Values are means ($n = 3$) $\pm$ 95% confidence interval.

absolute PRC coefficients revealed a strong treatment response for stages E ($4.1 \pm 0.9$) and F ($5.1 \pm 1.5$), compared to weak effects for stages A ($0.0002 \pm 0.0001$), B ($0.0005 \pm 0.0003$), C ($0.017 \pm 0.011$), and D ($0.27 \pm 0.14$).

### Larval density, vessel size, and fill volume

Both density (ANOVA, $F = 21.4, p < 0.005$) and vessel size (ANOVA, $F = 13.3, p < 0.005$) significantly influenced metamorphosis. The proportion of larvae reaching Stage E within 5 days was enhanced above 10 larvae mL$^{-1}$ for Petri dishes or 5 larvae mL$^{-1}$ for six- and twelve-well plates. Overall, the greatest proportion of Stage E larvae were observed in twelve-well plates (Fig. 3). Larval metamorphosis was enhanced when wells were filled to capacity ($t = 5.1, p = 0.007$). In wells filled to capacity, $87 \pm 5\%$ of larvae completed stage E of metamorphosis within 5 days compared to $45 \pm 5\%$ in control wells.

### Carrier solvent and reference compounds

No detectable effects on larval metamorphosis were observed when carrier solvent was added to the vessels at or below 20 µL mL$^{-1}$ for ethanol or 10 µL mL$^{-1}$ for DMSO. When dosed at 40 µL mL$^{-1}$, the number of larvae that reached stage E was reduced to zero in both cases (ethanol ANOVA: $F = 12.6, p < 0.005$; Dunnett's $p = 0.009$; DMSO ANOVA: $F = 16.0, p < 0.005$; Dunnett's $p = 0.02$).

Polygodial ($EC_{50} = 4.5$ ng mL$^{-1}$), portimine ($EC_{50} = 1.0$ ng mL$^{-1}$), oroidin ($EC_{50} = 1.1$ µg mL$^{-1}$), chlorothalonil ($EC_{50} = 0.1$ µg mL$^{-1}$), and tolylfluanid ($EC_{50} = 0.3$ µg mL$^{-1}$) potently inhibited metamorphosis; dose-response curves closely fit the data ($R^2 > 0.9$). The polygodial positive controls from the eight independent experiments had negative

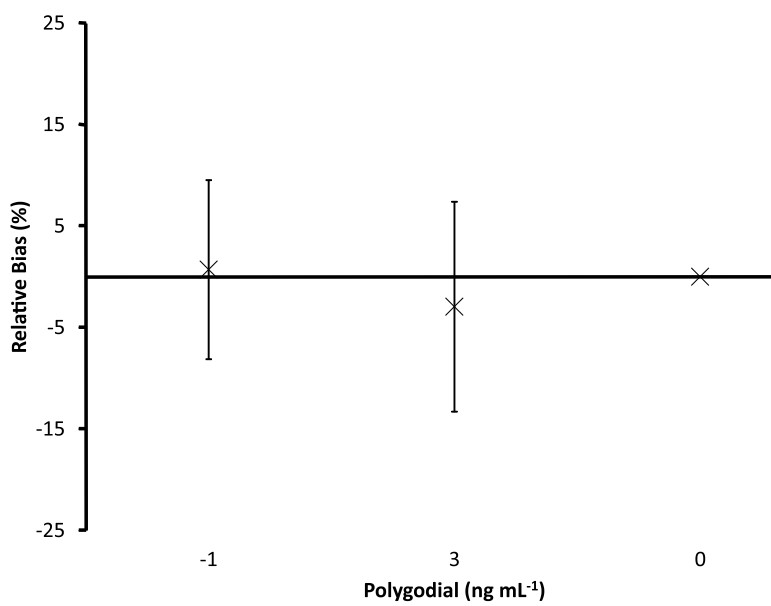

**Figure 4  Relative bias for polygodial treatments.** Relative bias from eight independent experiments for the effects of for polygodial treatments on *Ciona savignyi* metamorphosis. Values are means ($n = 8$) relative to a known dose-response relationship for polygodial $\pm$ 95% confidence interval.

**Table 1  Overall variability for polygodial treatments.** Variance component estimates and overall variability for the effects of polygodial treatments on *Ciona savignyi* metamorphosis.

|  | 5 ng mL$^{-1}$ | 10 ng mL$^{-1}$ | Mean |
| --- | --- | --- | --- |
| Var (Run) | −0.013 | 0.002 | −0.005 |
| Var (Error) | 0.033 | 0.030 | 0.031 |
| Overall | 15% | 20% | 17% |

dose-dependent effects on metamorphosis, with mean reductions in the proportions of larvae reaching stage E varying by $14 \pm 9\%$ within experiments (i.e., repeatability) and $17 \pm 3\%$ between experiments (i.e., intermediate precision; Fig. 4, Table 1). However, enrichment with an environmental extract decreased the potency of polygodial (Fig. 5). The extract alone had no detectable effect on metamorphosis but the observed shift in potency for enriched *vs.* pure polygodial represents an approximately 80% increase in $EC_{50}$ (7.8 vs. 4.3 ng mL$^{-1}$).

## DISCUSSION

The larvae of *C. savignyi* were well suited to laboratory culture and, as has been reported previously, they completed metamorphosis to established juveniles inside seven days (*Cirino et al., 2002*; *Hendrickson et al., 2004*). The proportion of larvae completing metamorphosis to stage E within five days exceeded 90% in some cases, but the performance of the larvae was highly dependent on environmental factors. Temperature and photoperiod strongly influenced metamorphic progress. Although no observable differences were found between

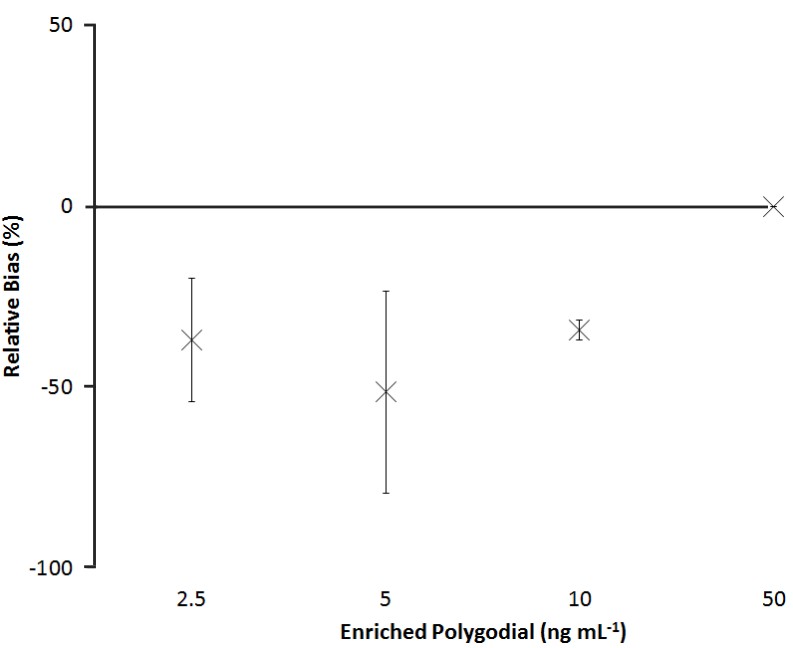

**Figure 5  Relative bias for polygodial enriched with an environmental extract.** Relative bias for the effects of polygodial enriched with an environmental extract on *Ciona savignyi* metamorphosis. Values are means ($n = 3$) relative to pure polygodial ±95% confidence interval.

15 and 21 °C, exceeding 21 °C effectively halted the progression of metamorphosis. A similar optimal temperature range of 12–20 °C has previously been reported for *C. savignyi* embryogenesis (*Nomaguchi et al., 1997*). Likewise, the observed enhancement of metamorphic progress for larvae reared under 12:12 h subdued light:dark fits the general rule that ascidians are initially positively phototactic but switch to being negatively phototactic when competent to settle (*McHenry, 2005*). It follows that free swimming larvae ought to prefer a lighted environment and, when competent to settle, will utilize the dark period to settle and begin metamorphosis.

Compared to temperature and photoperiod, substrate had only limited impact on *C. savignyi* larvae. Ascidians are generally considered to be one of the least discerning marine taxa in relation to surface selection (*Aldred & Clare, 2014*), yet there are anecdotal reports of surface selectivity in some species: *Ascidia nigra* Savigny settlement is negatively correlated with surface wettability (*Gerhart et al., 1992*), *Ascidia interrupta* Heller prefer sandblasted surfaces (*Rae Flores & Faulkes, 2008*), and *Ascidiella* spp. tend to be most abundant on concrete pilings (*Andersson et al., 2009*). Many other marine larvae will only attach to surfaces with defined chemical, biological, texture, or wettability characteristics (*Pawlik, 1992*; *Wahl, 1989*). For example, barnacle cyprids prefer smooth surfaces (*Berntsson et al., 2000*) and *Ulva* zoospores use quorum sensing signal molecules to seek out suitable biofilm communities (*Tait et al., 2005*). Granting that only a small number of simple substrate variations were tested and larvae were confined within small vessels, *C. savignyi* did not display obvious substrate preferences in the experiments performed here. The sides and bottom of the wells were of dissimilar composition, yet larvae adhered indiscriminately

to all surfaces. The exception was wells lined with smooth acrylic, where the overall progression of metamorphosis was reduced by approximately 12% compared to the other surfaces tested. This apparent insensitivity to surface characteristics may contribute to *C. savignyi's* invasive character, whereby larvae will settle on a wide range of available surfaces.

When the temperature, photoperiod, and substrate datasets were combined, the earliest developmental stage of metamorphosis to show strong treatment response was stage E when the branchial basket, siphons, and stigmata develop. It has previously been noted that tail-resorption is a particularly sensitive stage of ascidian metamorphosis (e.g., *Bishop, Bates & Brandhorst, 2001*; *Eri et al., 1999*; *Green et al., 2002*) but, in the case of *C. savignyi*, the tail resorption process (Stage B) was little affected by variations in temperature, photoperiod, or substrate. It is not clear why the later stages of metamorphosis were more sensitive to environmental variations but the lecithotrophic nature of *C. savignyi* may be partly responsible. Energy reserves within the larvae may become depleted later in metamorphosis (e.g., *Jaeckle, 1994*; *Jaeckle & Manahan, 1989*; *Moran & Manahan, 2003*), with energy deficient larvae becoming increasing susceptible to sub-optimal environmental conditions.

When larval density and vessel size were subsequently evaluated, the proportion of larvae reaching Stage E of metamorphosis was enhanced above 5 or 10 larvae mL$^{-1}$, with twelve-well plates having the greatest proportion of Stage E larvae overall. Density-dependent behaviours are common among marine larvae (*Hadfield & Paul, 2001*); examples include the gregariousness of larvae of the barnacle *Balanus amphitrite* Darwin (*Head et al., 2003*), the tubeworm *Hydroides dianthus* Verrill (*Toonen & Pawlik, 1996*), and the oyster *Ostrea edulis* L. (*Bayne, 1969*). Enhanced metamorphosis in smaller vessels could be a result of conspecific settlement cues or reduced surface area to volume ratio. Under the later scenario, larvae are more likely to contact, and thus adhere to, the internal surfaces of a smaller vessel.

Filling wells to capacity further increased the proportion of Stage E larvae observed after 5 days. When wells were not filled to capacity, some *C. savignyi* larvae became trapped at the air-water interface and where not considered to have successfully attached (P Cahill, pers. obs.). This phenomenon has been reported previously for other ascidians (e.g., *Fletcher & Forrest, 2011*), and is probably due to the larvae swimming upwards against the force of gravity for the first few hours after hatching (*Hendrickson et al., 2004*). Because culture waters contact the underside of the lid in wells that were filled to capacity, larvae are provided with an inverted surface upon which to settle and are prevented from them becoming trapped at the air-water interface. In line with these findings, it has been anecdotally observed that *C. savignyi* adults tend to be congregated on the underside of man-made structures in the sea (P Cahill, pers. obs.).

Evaluating the effects of select reference compounds on *C. savignyi* metamorphosis highlighted the potential to use *C. savignyi* larvae to screen for bioactivity under an optimised laboratory setting. The natural antifouling agent polygodial (*Cahill & Kuhajek, 2014*),the algal biotoxin portimine (*Selwood et al., 2013*), and the synthetic antifouling biocides chlorothalonil and tolyfluanid (*Voulvoulis, Scrimshaw & Lester, 1999*) were

bioactive in the low ng mL$^{-1}$ to high μg mL$^{-1}$ ranges. Tolerance of *C. savignyi* larvae for the carrier solvents ethanol (20 μL mL$^{-1}$ max.) and DMSO (10 μL mL$^{-1}$ max.) facilitated screening these compounds, which, with the exception of portimine, have only limited water solubility. The results reported here meet or exceed potency estimates previously determined for these compounds against other organisms. For example, polygodial is effective against fungal pathogens at approximately 1 μg mL$^{-1}$ (*Kubo & Himejima, 1991*), chlorothalonil kills water fleas and fathead minnows at 0.03–0.2 μg mL$^{-1}$ (*Sherrard et al., 2002*), and tolylfluanid controls seaweed zoospores at 0.03 μg mL$^{-1}$ (*Wendt et al., 2013*). However, systems that test the bioactivity of compounds in solution must be carefully interpreted. In nature, marine larvae are typically only exposed to allelopathic metabolites at surfaces, such as when they contact another benthic organisms (*Pawlik, 1993*). Perfusing larval tissues with a compound, as was done here, provides a largely pharmacological rather than ecological context. This pharmacological data is often used as a first step to identify promising bioactive compounds but should always be followed by additional investigations, either laboratory or field based, where the compound of interest is bound to/released from a surface (*Bressy et al., 2014*).

When polygodial treatments were included in eight independent experiments performed over a one year period, mean reductions in the proportion of larvae reaching stage E varied by 14 ± 9% and 17 ± 3% within and between experiments, respectively. These values represent a relatively high degree of consistency for the effects of polygodial exposure, with comparable larval systems returning repeatability estimates in the order of 5–30% (e.g., *Piazza et al., 2012*; *Ross & Bidwell, 2001*; *Stronkhorst et al., 2004*). The larvae were unaffected by a complex environmental extract produced from the culture waters of green-lipped mussels but the same extract reduced the potency of polygodial to almost half. This result highlights the potential for changes in potency when working with complex chemical samples, as is common when attempting to isolate natural products (*Colegate & Molyneux, 2007*).

Overall, this study yielded insights into the environmental factors impacting metamorphosis of *C. savignyi* larvae. This increased understanding of the interplay between the environment and the biology of *C. savignyi* is vital to understanding, and perhaps counteracting, the invasive tendencies of this organism. The findings also provide context for the growing body of research examining the biochemical and genetic determinants of *C. savignyi* metamorphosis. Demonstrating that *C. savignyi* larvae are sensitive to a range of natural and synthetic bioactive agents highlighted one potential application for these larvae. Using *C. savignyi* larvae to assess the toxicity of compounds under optimised laboratory conditions presents a potentially useful preliminary screening tool but the outcomes of such experiments must be carefully interpreted and should be followed by ecologically relevant investigations. Nevertheless, chemicals targeting *C. savignyi* metamorphosis could potentially be used to control this pest species.

### Funding
This study was funded by the New Zealand Ministry of Business Innovation and Employment (CAWX1315). The funders had no role in study design, data collection and analysis, decision to publish, or preparation of the manuscript.

### Grant Disclosures
The following grant information was disclosed by the authors:
New Zealand Ministry of Business Innovation and Employment: CAWX1315.

### Competing Interests
The authors declare there are no competing interests.

### Author Contributions
- Patrick L. Cahill conceived and designed the experiments, performed the experiments, analyzed the data, wrote the paper, prepared figures and/or tables.
- Javier Atalah analyzed the data, reviewed drafts of the paper.
- Andrew I. Selwood conceived and designed the experiments, contributed reagents/materials/analysis tools, reviewed drafts of the paper.
- Jeanne M. Kuhajek conceived and designed the experiments, reviewed drafts of the paper.

### Data Availability
   The raw data has been supplied as Data S1.

### Supplemental Information
Supplemental information for this article can be found online at http://dx.doi.org/10.7717/peerj.1739#supplemental-information.

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
