# Peer review of "Metamorphosis of the invasive ascidian Ciona savignyi: environmental variables and chemical exposure"

_PeerJ, doi:10.7717/peerj.1739_

## Round 0.1 · original submission · Major Revisions

Both reviewers of this ms. found it to be a satisfactory contribution to our understanding of ascidian settlement and metamorphosis, but disagreed that it represents an advance as an antifouling bioassay. I agree with this assessment. The literature reviews cited in this paper (some of which I authored) discuss at length the ecological relevance of settlement bioassays – specifically, that larvae in nature are not exposed to high concentrations of metabolites that are found inside the tissues of benthic organisms, and are likely to respond negatively (or positively) only on contact with surfaces. Assays that perfuse larval tissues in this manner are more pharmacological than ecological. Commercially useful antifouling compounds would similarly be most cost effective as surface-bound agents. These issues are inadequately addressed in the ms.

Major revisions are recommended for this ms., specifically re-casting it as a contribution to understanding ascidian larval responses to various treatments under laboratory conditions, with clear context related to the limitations of lab assays of pure or extracted compounds in solution.

Reviewer 1 ·

Basic reporting

Fail
This article falls short in reporting of previously published work, especially on ascidian bioassays which have been reported for about 3 decades. Ciona intestinal and Acidia nigra come to mind. If the assay is purported to be one that is to be used with chemistry, then speed of assessment is particularly important. Previous assays go at most to tail resolution because this is a rapid endpoint. Running an assay for 5 days with a single addition of a chemical makes no sense unless the goal is molecules that are very stable. If one thinks about this, very stable molecules, when one gains market share, build up in environments. it would be good to look at the literature on assays used to do bioassay directed purification and assays to assess antifouling activity of natural products and examine their features in relation to your assay.

Experimental design

Fail,
other assays have used ascidians and have tested different surface energies and found very high specificity in responses. The plastics and surface energies used in these studies, which are not expensive and easy to obtain were not tested.

Validity of the findings

Did a very good job on this. The data are fine, the premise of an antifouling bioassay is the one that is not well supported.

Additional comments

I've made my major comments in the first two boxes above. It is really important if you want to propose a bioassay for studying antifouling compounds that you propose an assay that is fast and relevant to the topic. Watching development to a fully developed ascidian is not the best idea.

Reviewer 2 ·

Basic reporting

The authors claim to have developed a ‘simple’ bioassay for screening compounds and natural product extracts for antifouling activity against ascidians. However, the bioassay described here confounds toxicity and antifouling effects (i.e., for a meaningful antifouling assay, compounds or natural extracts should be incorporated in an adequate matrix that would release them at a meaningful rate and where the larvae could potentially settle, not directly released in the well water as the authors seem to have done here). In addition, Ciona savignyi larvae are not readily available further undermining the usefulness of the assay described here.

Experimental design

The authors were very thorough with their testing and considered many variables that could influence larval settlement, including the photoperiod, substrate type, and temperature. Larval density, well size, and carrier solvents were also investigated. However, as mentioned above, the described experimental setting is not adequate to test for antifouling properties of natural products or extracts.

Validity of the findings

There are few studies that have looked into which parameters may influence ascidian larval development and survival and thus results described here are very interesting. Assays were well designed and very thorough and results were correctly analyzed and interpreted. However, these findings are not adequate to validate an antifouling assay, which unfortunately was the main goal of this manuscript.

Additional comments

Thorough the manuscript, the authors confound ‘larval development’ with ‘metamorphosis’.
Abstract: Lines 30-31. Add average value for repeatability and intermediate precision values followed by their standard deviation or standard error values.
Introduction. Please, add further information about C. savignyi biological cycle. How often does this species reproduce? How fast does it grow? What is its actual distribution range? How easy would it be to obtain specimens of this species and keep them alive in aquaria? Etc.
Material and methods. Lines 100-101. What do you mean by ‘substrate types were varied individually’?
Material and Methods. Lines 160-169. Please, explain why polygodial was enriched with an ‘environmental extract’ and why that specific environmental extract (containing mussels) was selected.
Results. Lines 227-229. Indicate which stages were most affected by these compounds.
Results. Lines 229-232. Please, consider other repeatability and intermediate precision indicators. ±14 and 17% is too vague (e.g., ±14% of what exactly?).
Please, update and add relevant references throughout the manuscript.

---

## Round 0.2 · accepted · Accept

The authors have done a good job of re-casting this ms. as a contribution to our understanding of ascidian settlement and metamorphosis under lab conditions and in response to specific natural products.